# Hierarchical System for Recognition of Traffic Signs Based on Segmentation of Their Images

Sergey Victorovich Belim [1,2,*], Svetlana Yuryevna Belim [1] and Evgeniy Victorovich Khiryanov [2]

1    Radio Engineering Department, Omsk State Technical University, Omsk 644050, Russia; syubelim@omgtu.ru
2    Department of Digital Technology, Siberian State Automobile and Highway University, Omsk 644080, Russia; hiryanove@mail.ru
*    Correspondence: sbelim@mail.ru

**Abstract:** This article proposes an algorithm for recognizing road signs based on a determination of their color and shape. It first searches for the edge segment of the road sign. The boundary curve of the road sign is defined by the boundary of the edge segment. Approximating the boundaries of a road sign reveals its shape. The hierarchical road sign recognition system forms classes in the form of a sign. Six classes are at the first level. Two classes contain only one road sign. Signs are classified by the color of the edge segment at the second level of the hierarchy. The image inside the edge segment is cut at the third level of the hierarchy. The sign is then identified based on a comparison of the pattern. A computer experiment was carried out on two collections of road signs. The proposed algorithm has a high operating speed and a low percentage of errors.

**Keywords:** image recognition; traffic signs; classification methods; image segmentation

## 1. Introduction

Traffic sign recognition algorithms in an image are part of many technical computer vision systems. Technical systems have different requirements for the properties of these algorithms. Methods of comparing algorithms are based on two main properties: speed and the number of resources. Automatic mobile objects require a short operating time for the algorithm. Automatic road inventory systems impose less stringent requirements. Video analysis systems from cars are the least demanding. The system described in this article is an integral part of automatic road inventory software.

There are three types of image processing algorithms that contain traffic signs. The first type of algorithm detects traffic signs in an image of a road scene [1,2]. The second type of algorithm classifies road signs by their localized image [3,4]. The third type of algorithm includes the features of the first two. These algorithms detect traffic signs based on attempts to classify fragments of road scenes [5,6]. The third type of algorithm requires the most computational resources. Most vision systems separate the problems of detecting and classifying traffic signs. In this article, we consider only the problem of classifying traffic signs. A localized traffic sign image is input into the algorithm. There are several collections of such images for testing algorithms. The most widespread is the German Traffic Sign Recognition Benchmark (GTSRB) [7].

The main achievements in the recognition of traffic signs from images are related to the use of neural networks. The recognition of road signs boils down to their classification. A multi-column neural network recognition and image preprocessing system [8] shows a high efficiency (99.46%) in the GTSRB collection. This system uses 25 different neural networks. Each neural network has its own training data set. The system performance is low. The system is slow due to the large number of parameters (90 million). Preliminary spatial transformations reduce the number of neural network parameters. The authors of [9,10] reduced the number of parameters to 14 million. The architecture of such a neural network has three layers. Each layer performs spatial transformations. These operations





eliminate background and geometric noise. The system shows an efficiency of 99.71% in in the GTSRB collection. Another approach builds a neural network not in layers, but on the basis of a hierarchical tree [11]. The number of parameters for such a network is 6.256 million. The recognition efficiency remains high (99.33%) in the GTSRB collection. A hierarchical neural network can be built on the basis of dividing many traffic signs into subsets [12]. The neural network is trained separately for each subset. The effectiveness of this method is 99.67% in the GTSRB collection. An expert must perform a subset split before this algorithm can work. The learning process also has an impact on the number of neural network parameters. The authors of [13] learned the "teacher" neural network. This network has 7.4 million parameters. The "teacher" neural network learns the "student" neural network. This neural network has 0.8 million parameters. The "student" neural network recognizes traffic signs. The efficiency of this two-stage system is 99.61% in the GTSRB collection. This approach can lead to divergence during the transition between neural networks. The system is very sensitive to insufficient training of the "teacher" neural network. In this case, the efficiency of work is greatly reduced.

There are several problems when using neural networks to recognize traffic signs: the need for high performance, low efficiency in real time, and a large amount of input data [14]. Sensitivity to uncontrolled environments is also a characteristic of neural-network-based traffic sign recognition systems [15]. The problem of reducing our dependence on neural networks is urgent. One solution is to split an image of a road sign into parts using a segmentation algorithm. Segmentation algorithms are used primarily to detect traffic signs. Traffic sign detection methods cluster a road scene into one of the following color spaces: RGB [16], HSV [17], and YUV [4].

The results obtained for the neural network approach can be used to build classifiers without using neural networks. Hierarchical representation for traffic sign sets reduces the number of classifier parameters. Splitting an image of a traffic sign into segments speeds up the classification process.

This article suggests a hierarchical algorithm for the recognition of traffic signs based on the segmentation of an image and a comparison with standards.

## 2. Hierarchical Classification of Traffic Sign Images

One of the main problems in the classification of traffic signs is the large number of their standard types. The total number of standard types for traffic signs is more than one hundred. Classifier efficiency is improved by reducing the number of object classes. The classification of traffic signs can be carried out by the method of sequential approximation. The classifier sequentially clarifies the information about a traffic sign by its image. The number of classes at each stage must be small. The shape and color of a traffic sign is specified in the first two stages. The details of the internal image are specified in the third step. The number of stages and their content differ for different types of traffic signs. The hierarchical system of traffic sign recognition methods corresponds to such a task. There is a hierarchy of traffic signs according to their effect on road traffic. This hierarchy cannot be used in a traffic sign recognition system. There is no complete correlation between the shape and color of a traffic sign and its value.

A zero-level classification is related to the geometric shape of the signs. The first class at this level includes traffic signs that can be inscribed in a square (square, round, triangular, octagonal, rhombic). These traffic signs perform prescriptive and prohibitive functions. The second grade at level zero includes rectangular-shaped traffic signs. These signs inform drivers. Our classifier only works with zero-level first-class signs. Traffic signs in this class are critical to traffic management. Our classifier can easily be extended to work with second-class signs. This task requires the development of additional templates.

A hierarchical road sign recognition system requires the development of classifiers for each level of the hierarchical tree. The classifiers may be the same for the type of algorithms used. However, each classifier requires its own training set.

A first-level classifier recognizes the shape of a traffic sign. We denote the first-level classifier as $Z$. Classifier $Z$ distributes traffic signs into six classes as follows: $z1$—rhombus, $z2$—octagon, $z3$—inverted triangle, $z4$—circle, $z5$—square, and $z6$—triangle. The first level for the hierarchical classifier tree is shown in Figure 1.

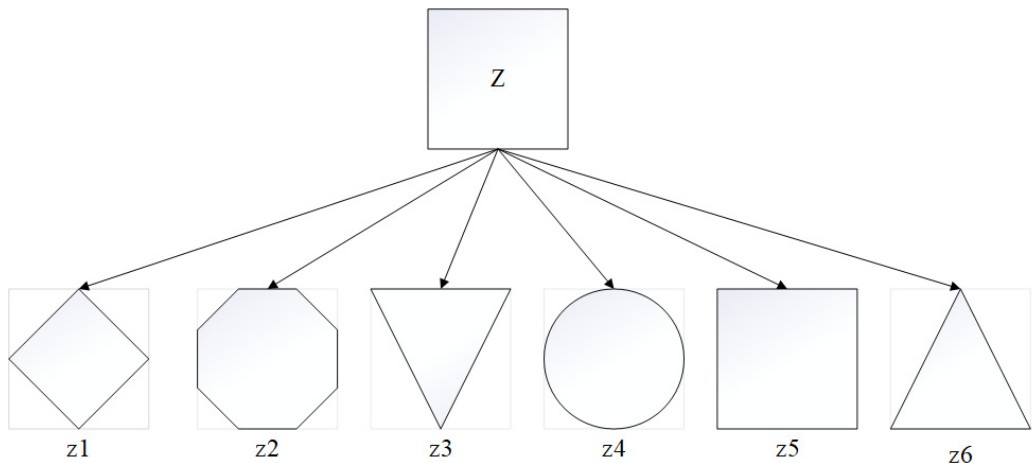

**Figure 1.** First level of traffic sign classification hierarchy. $Z$ is the first-level classifier.

Some second-level classes do not contain the next-level classifiers. Classes $z2$ and $z3$ include one traffic sign each (Figure 2). These nodes are the leaves in a hierarchical tree. If the classifier $Z$ assigns a traffic sign to one of these classes, then the recognition problem is solved.

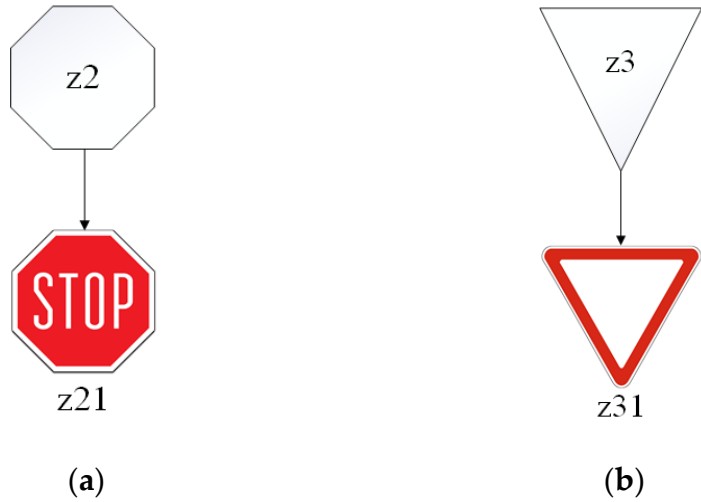

(**a**)                                                              (**b**)

**Figure 2.** Classes that include one traffic sign: (**a**) $z2$; (**b**) $z3$.

The class $z1$ contains only two traffic signs (Figure 3). The identification of class $z1$ traffic signs requires one additional step. Recognition in this class can also be performed based on character outline highlighting.

The class $z4$ includes five subclasses (Figure 4). Subclasses $z41$ and $z45$ can be separated from the others with red along the contour. Subclass $z45$ differs from subclass $z41$ in the number of pixels in red. Node $z45$ is a leaf. If a road sign is assigned to this class, then the recognition problem is solved. Class $z41$ includes 32 traffic signs. These signs have a different image of black on a white background. Classes $z43$ and $z44$ are highlighted by the presence of an area of blue. Class $z44$ is separated from class $z43$ by a red stripe. Class $z42$ includes nine traffic signs. Seven traffic signs in this class contain images of

numbers. Class $z43$ includes 18 traffic signs. These traffic signs contain a white image on a blue background.

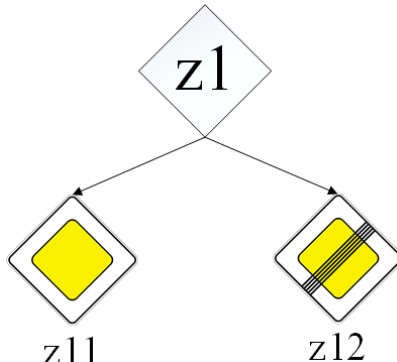

**Figure 3.** Class z1 with two traffic signs.

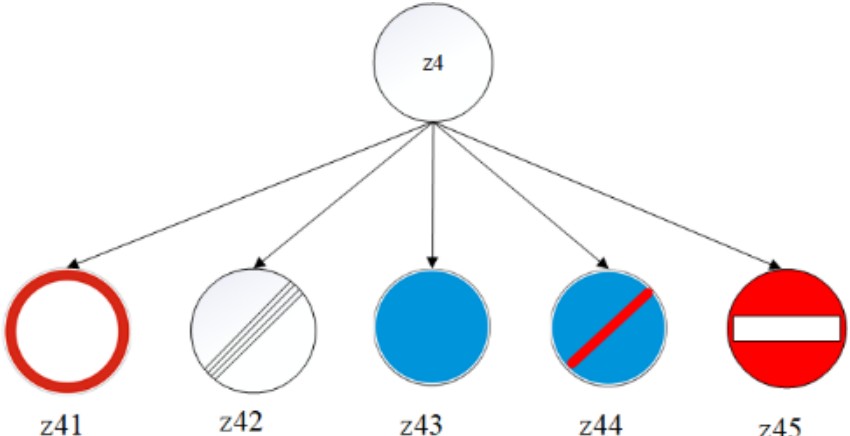

**Figure 4.** Subclasses of class $z4$.

An additional hierarchy level in class $z4$ simplifies the classification task. We introduce three additional classes (Figure 5). The $z4(1)$ class combines subclasses $z41$ and $z45$. The $z4(2)$ class includes only subclass $z42$. The $z4(3)$ class combines subclasses $z43$ and $z44$.

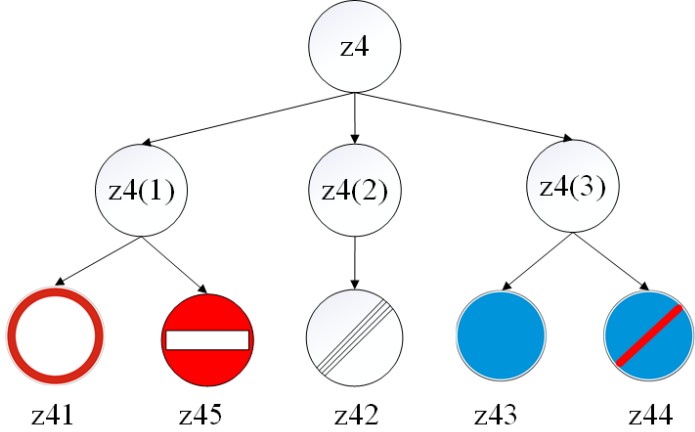

**Figure 5.** Additional hierarchy level in class $z4$.

An edge segment is a segment of a traffic sign image that includes its outer boundary. Traffic signs in class $z4(1)$ have a red edge segment. Traffic signs in class $z4(2)$ have a white edge segment. Traffic signs of class $z4(3)$ have a blue edge segment.

Splitting the $z4(1)$ class into two classes, $z41$ and $z45$, is easy. The area of the edge segment in class $z45$ is more than twice the area of the edge segment in class $z41$. Class $z45$ is a leaf.

The inner segment of traffic signs in class $z41$ is highlighted by removing the edge segment and the entire image outside the sign from the image. The inside of the sign is a black and white image without grayscale. A binarization operation is applied to the inner segment of the traffic sign. The binary shape of the image is easily compared to the pattern.

Class $z5$ includes three subclasses (Figure 6). The main difference between these classes is the colors used. The subclass $z51$ includes six traffic signs. These signs define the direction of the main road. The inner segment of these classes consists of black lines. Subclass $z52$ contains 33 traffic signs. Subclass $z53$ contains seven traffic signs.

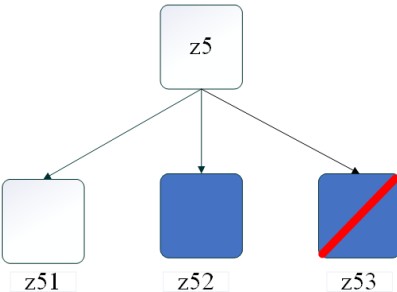

**Figure 6.** Class $z5$ subclasses.

The edge segment type requires an additional hierarchy level for class $z5$. The first subclass $z5(1)$ contains traffic signs with a white edge segment. The second subclass of $z5(2)$ includes traffic signs with a blue edge segment (Figure 7). The RGB model makes it easy to distinguish between these classes.

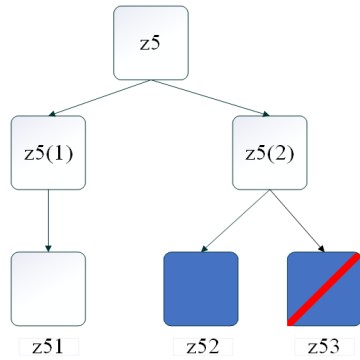

**Figure 7.** Additional hierarchy level in class $z5$.

All traffic signs in class $z6$ have the same shape and edge segment (Figure 8). Class $z61$ includes 42 traffic signs.

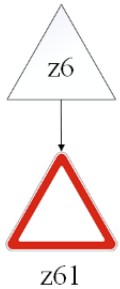

**Figure 8.** The class $z6$.

The color of the edge segment in class z61 does not affect the recognition of the traffic sign. The algorithm shall determine the shape of the traffic sign. The edge segment and the entire exterior environment are removed from the image. The inner segment of the road sign is a black and white image without grayscale. The contents of an internal segment can be defined based on a comparison with a template.

An analysis of the image hierarchy for traffic signs shows that the recognition of traffic signs based on their segmentation includes five algorithms:

1. Algorithm for obtaining an edge segment and determining its color (red, blue, white);
2. Algorithm for determining the area of the edge segment;
3. Algorithm for determining the shape of a traffic sign by its edge segment;
4. Algorithm for obtaining an inner segment of a traffic sign and its binarization;
5. Algorithm for identifying the internal image of a traffic sign.

## 3. Segmentation Algorithm

The edge segment is searched for a localized traffic sign. We do not solve the problem of localizing the traffic sign on the road scene. Various algorithms can be used to localize a traffic sign in a square window. There are localization algorithms for similar tasks [18,19]. These algorithms can be adapted for traffic signs. The task of finding traffic signs in an image requires separate research. A square window including a traffic sign is a window for localizing a traffic sign.

We use the segmentation algorithm developed earlier to highlight the road contour [20–22]. This segmentation algorithm highlights segments that have both a uniform color fill and a gradient color fill. This factor is important in the processing of real traffic signs. The illumination of the sign is never uniform. This results in color gradients in images of real traffic signs. Our algorithm has a quadratic complexity. The segmentation algorithm is based on the method of growing areas. The method of growing areas has a quadratic labor intensity. Each iteration of the algorithm performs a number of operations not exceeding the number of image points. The total number of iterations does not exceed the number of image points. The speed of the algorithm is large enough for use in the designed system. The size of the input images of the algorithm does not exceed $200 \times 200$ pixels. The segmentation target in this case is the edge segment of the traffic sign. An edge segment is a segment of a traffic sign image having an outer boundary coinciding with the boundary of a traffic sign. All points of this segment have the same color in the ideal image of the traffic sign. Real images distort the color of individual pixels due to lighting features. Foreign objects and shadows change the outer boundary of the edge segment. These factors complicate the segmentation task. We enter the symbol $C$ for the set of edge segment points.

The segmentation algorithm is based on the method of growing areas. The grain of the algorithm is one pixel. The dimensions of the traffic sign localization window are $N \times N$ pixels. We select the starting pixel of the segment at $p_0 = (N/2, b)$. The coordinates of this point are determined from the upper-left corner. The value of the first coordinate for point $p_0$ is based on the symmetry of all traffic signs with respect to the vertical axis through their center. The traffic sign localization algorithm determines the second coordinate $b$. The value of this coordinate depends on the accuracy of finding the signs' boundaries. This coordinate should not have a large value, otherwise the selection of a narrow red strip of prohibitory traffic signs is impossible. $b = 0.1N$ must be taken for the GTSRB collection.

An undirected weighted graph can be mapped to each image. The nodes of this graph are the pixels of the image. $V$ is the set of the graph nodes. The edges of the graph connect the nodes. We consider a fully connected graph at the beginning of the algorithm's construction. The edge weight determines the color difference of the image pixels corresponding to the graph nodes. The RGB model is used for representing the color of the pixels. Each pixel has five coordinates. Two coordinates $(x, y)$ determine its position in the image. Three coordinates $(r(x, y), g(x, y), b(x, y))$ determine its color characteristics. $r(x, y)$ is the intensity of red. $g(x, y)$ is the intensity of green. $b(x, y)$ is the intensity of blue.

We define the distance in terms of the five-dimensional space $d(p, p')$ between the pixels $p = (x, y, r, g, b)$ and $p' = (x', y', r', g', b')$.

$$d(p, p') = \sqrt{(x - x')^2 + (y - y')^2}\left(exp\left(\sqrt{(r - r')^2 + (g - g')^2 + (b - b')^2}\right) - 1\right). \quad (1)$$

The five-dimensional distance between the pixels increases if the distance between the pixels in the image grows or the difference in color increases.

The image segmentation problem is equivalent to partitioning a graph into communities [23]. A community must include the nodes of the graph whose pixels are located a short distance from each other and have a similar color. Having a small distance is a necessary requirement for segment connectivity. The similar colors ensure the edge segment is highlighted. The edge segment of the traffic sign has one color. The edge weight of the graph should decrease when moving away from the pixel and when the color changes sharply. The edge weight is calculated based on the distance between pixels $p$ and $p'$. If node $v$ corresponds to pixel $p$ and node $v'$ corresponds to pixel $p'$, then the weight of the arc between nodes $v$ and $v'$ is calculated using the following formula:

$$w(v, v') = \frac{1}{1 + d(p, p')}. \quad (2)$$

This function for the edge weight is 1 if the neighboring pixels and their pixel colors match. If the neighboring pixels have different colors, then $w(v, v') < 1$. The function $w(v, v')$ rapidly decreases as the distance between pixels increases. Accounting for the edges between nodes for pixels very far from each other is not necessary. We enter the correlation radius $R$. This value indicates the largest geometric distance between pixels whose nodes are connected by an edge. The edges between nodes that are located in the image more than $R$ are removed from the graph. The correlation radius determines the number of neighbors for a node.

A community search is required to highlight a segment. We use the method of growing regions on a graph. Community $H$ with strongly connected nodes is the result of the algorithm. Community $H$ includes one node $v_0$ at the zero step of the algorithm. This node $v_0$ corresponds to pixel $p_0$. The algorithm at each stage calculates the average edge weight in community $H$.

$$w(H) = \frac{1}{n} \sum_{v, v' \in H} w(v, v'). \quad (3)$$

$n$ is the number of nodes in community $H$.

The algorithm step is to enumerate all nodes connected by an edge to at least one community $H$ node. If node $v_i$ is a candidate for inclusion in community $H$, then the total association of this node with nodes of community $H$ is as follows:

$$w_i = \sum_{v \in H} w(v_i, v). \quad (4)$$

If the $w_i$ does not exceed the threshold $w_0(H)$, then node $v_i$ is included in community $H$. The threshold value $w_0(H)$ is calculated based on the average weight of the community edges.

$$w_0(H) = hw(H). \quad (5)$$

The parameter $h$ is determined in a computer experiment. This parameter depends on the type of images. The algorithm achieves the best results for traffic signs at $h = 0.07$.

The algorithm stops if no node satisfies the condition of inclusion in community $H$. The complexity of the algorithm depends on the number of pixels in the image according to the quadratic law. If the traffic sign localization window has a size of $N \times N$ pixels, then the complexity of the algorithm grows according to the law $O(N^4)$. The edge segment $C$ consists of pixels corresponding to nodes in community $H$.

The outer boundary of the edge segment is the boundary of the traffic sign. The edge segment boundary algorithm is reduced to determine the number of neighbors for each pixel in segment *C*. If a pixel in segment *C* has less than eight neighbors from that segment, then it is boundary. The outer edge of the edge segment is separated from the inner edge by the geometric coordinates of the points. The sign boundary in this algorithm has a thickness of one pixel. Examples of edge segments and boundary definition for traffic signs are shown in Figure 9.

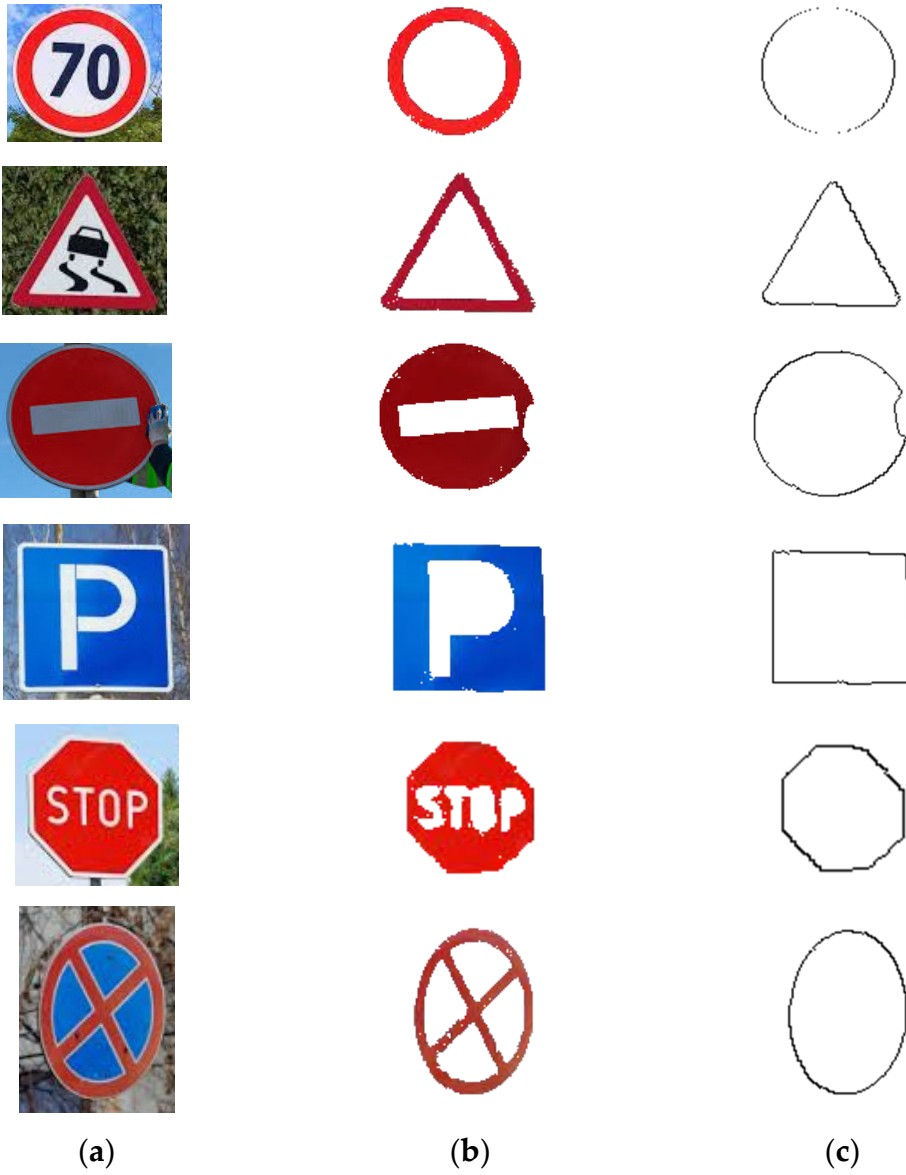

(**a**)          (**b**)          (**c**)

**Figure 9.** Examples of edge segments and boundary definition for traffic signs: (**a**) traffic sign images, (**b**) edge segment of the traffic sign, and (**c**) traffic sign boundary.

The color of the edge segment is determined by comparing the average value of the color coordinates for its pixels. There are three colors of the edge segment (red, blue, white). We calculate the average value of the red and blue components for pixels in the edge segment.

$$R(C) = \frac{1}{K} \sum_{p \in C} r(p), B(C) = \frac{1}{K} \sum_{p \in C} b(p). \tag{6}$$

$K$ is the number of pixels in the edge segment $C$. $r(p)$ is the intensity of red for pixel $p$. $b(p)$ is the intensity of blue for pixel $p$. The mean difference determines the color of the edge segment.

$$\Delta C = R(C) - B(C). \tag{6}$$

We compare the difference in average values $\Delta C$ with the depth of palette $M$. If $\Delta C > M/3$, the edge segment is red. If $\Delta C < -M/3$, the edge segment is blue. If $M/3 < \Delta C < M/3$, the edge segment is white. In this case, the intensities of red and blue are approximately equal. The green component of the pixels is not considered, since there are no green road signs.

## 4. Recognition of the Traffic Sign Shape

We use templates to recognize the shape of a traffic sign. There are six different templates: rhombus ($t_1$), octagon ($t_2$), inverted triangle ($t_3$), circle ($t_4$), square ($t_5$), and triangle ($t_6$). The traffic sign outline is mapped to all the templates. The decision on the shape of the sign is made as closely as possible to one of the templates. There are three main problems when comparing the outline of a traffic sign with templates. The first problem is the variability in the dimensions of the sign in the image. The solution to this problem is templates scaling. The second problem is the distortion of the traffic sign when shooting at an angle. The solution to this problem is to use deformable templates. The two main characteristics of the template are the number of angles and the location on the plane. The third problem is the distortion of straight lines when obtaining the contour of a traffic sign. The features of real images prevent smooth lines and curves.

The first step in determining the shape of a traffic sign is to calculate its center. All geometric figures of traffic signs have central symmetry. For all traffic sign shapes, the center of symmetry coincides with the center of mass. We enter the designation $D$ for the set of points for the traffic sign outline obtained using the segmentation algorithm. Formulas for the center of mass $(x_c, y_c)$ for the set of points with the same mass are used when finding the center of the traffic sign image.

$$x_c = \frac{1}{M} \sum_{(x_i, y_i) \in D} x_i, y_c = \frac{1}{M} \sum_{(x_i, y_i) \in D} y_i. \tag{7}$$

$M$ is the number of points in set $D$.

Each template $t_j$ ($j = 1, 2, 3, 4, 5, 6$) is described by an equation.

$$P_j(x, y) = 0. (j = 1, 2, 3, 4, 5, 6) \tag{8}$$

The points of set $D$ do not satisfy any template equation. The deviation $e_j$ of the coordinate for the points of set $D$ from the template equation $P_j(x, y)$ determines the proximity of the shape to the template $t_j$.

$$e_j = \frac{1}{M} \sqrt{\sum_{(x_i, y_i) \in D} \left(\Delta P_j(x_i, y_i)\right)^2}. (j = 1, 2, 3, 4, 5, 6) \tag{9}$$

$\Delta P_j(x_i, y_i)$ is the deviation at the location of point $(x_i, y_i)$ from the curve for the template $P_j(x, y)$. The minimum $e_j$ defines the template $t_j$ to which the sign belongs.

Next, we consider the features of creating adaptable templates for various shapes.

### 4.1. Circle Template ($t_4$)

A circular traffic sign is very rarely circle-shaped in a real image. Angled shooting turns the circular sign into an ellipse in the image. The main axes of the ellipse can be oriented arbitrarily. Analyses of various images for traffic signs have shown that when

shooting from a car, the positions of the ellipse axes are close to vertical and horizontal. We use the ellipse equation in this case.

$$P_4(x,y) = \frac{x^2}{a^2} + \frac{y^2}{b^2} - 1. \tag{10}$$

The lengths of the major semi-axes $a$ and $b$ of the ellipse are key information in this equation. The algorithm determines the point of set $D$ furthest from the center of the ellipse $(x_c, y_c)$. The distance from this point to the center is $b$ if the horizontal coordinate of the point is close to $x_c$; otherwise, it is $a$. After that, the algorithm is the closest point to the center of the ellipse. The distance from this point to the center defines the second semi-axis. Substituting these values into Formula (10) gives us the $t_4$ template equation for an ellipse $P_4(x,y)$.

The deviation of location for the points $(x_i, y_i)$ from a template $\Delta P_4(x_i, y_i)$ is calculated by the substitution of the coordinates for this point in the ellipse equation $P_4(x,y)$.

$$\Delta P_4(x_i, y_i) = P_4(x_i, y_i). \tag{11}$$

### 4.2. Triangle Template ($t_3$, $t_6$)

The two templates for triangular traffic signs $t_3$ and $t_6$ differ in the mutual arrangement of the vertices. The two vertices of the triangle have a close coordinate $y$ value. The third vertex has a smaller coordinate $y$ value for the template $t_6$. The third vertex has a larger coordinate $y$ value for the template $t_3$.

The first step of the algorithm determines the coordinates of the vertices for the triangle $(x_1, y_1)$, $(x_2, y_2)$, $(x_3, y_3)$, and $(x_4, y_4)$. The traffic sign localization window is divided into six areas (Figure 10).

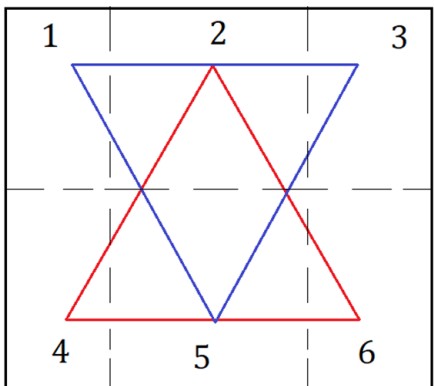

**Figure 10.** Splits the traffic sign localization window to define the vertices of the triangle.

The algorithm determines a point of set D with the maximum distance from the center of the traffic sign contour $(x_c, y_c)$. The algorithm searches the locality of the point $(0,0)$. This point is the first vertex of the triangle $(x_1, y_1)$. If the first vertex of the triangle is located in region 1, then the algorithm looks for two other vertices in regions 3 and 5. This vertex location corresponds to the template $t_6$. If the first vertex of the triangle is located in region 2, then the algorithm looks for two other vertices in regions 4 and 6. This vertex location corresponds to the template $t_3$. The points most distant from the center of the traffic sign in the desired area correspond to the vertices of the triangle.

Triangle side equations are necessary to calculate the deviation of points from the template. The equation of a line is written through the coordinates of the two vertices lying on that line. The distances to these three lines are calculated for each point $(x_i, y_i)$ of contour $D$.

$$d_1(x_i, y_i) = \frac{|(y_1-y_2)(x_i-x_1)+(y_i-y_1)(x_1-x_2)|}{\sqrt{(y_1-y_2)^2+(x_1-x_2)^2}},$$
$$d_2(x_i, y_i) = \frac{|(y_1-y_3)(x_i-x_1)+(y_i-y_1)(x_1-x_3)|}{\sqrt{(y_1-y_3)^2+(x_1-x_3)^2}}, \qquad (12)$$
$$d_3(x_i, y_i) = \frac{|(y_2-y_3)(x_i-x_2)+(y_i-y_2)(x_2-x_3)|}{\sqrt{(y_2-y_3)^2+(x_2-x_3)^2}}.$$

The contour point must be mapped to the nearest side of the triangular template. The deviation from the template for point $(x_i, y_i)$ is equal to the distance to the nearest side of the template.

$$\Delta P_j(x_i, y_i) = \min(d_1(x_i, y_i), d_2(x_i, y_i), d_3(x_i, y_i)).(j = 3, 6) \qquad (13)$$

### 4.3. Quadrangle Template ($t_1, t_5$)

The two types of quadrangular templates $t_1$ and $t_5$ differ from each other in the mutual arrangement of vertices. For a rhombus $t_1$, two vertices have the same $y$ coordinates. The other two vertices have a coordinate $y$ greater than and less than this. A square has two pairs of vertices with the same $y$ coordinate. The first step of the algorithm determines the coordinates of the vertices for the quadrangle $(x_1, y_1)$, $(x_2, y_2)$, $(x_3, y_3)$, and $(x_4, y_4)$. The traffic sign localization window is divided into nine areas (Figure 11).

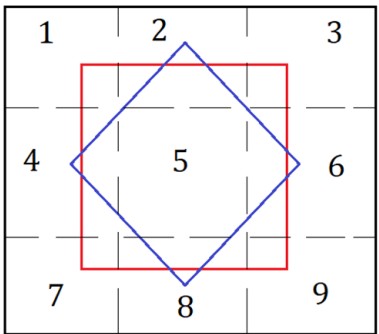

**Figure 11.** Splits the traffic sign localization window to define the vertices of the quadrangle.

The algorithm determines a point from the set $D$ with the maximum distance from the center of the traffic sign contour $(x_c, y_c)$. This point is searched in the vicinity of the origin $(0,0)$. The point found is the first vertex of the quadrangle $(x_1, y_1)$. If the first vertex of the quadrangle is located in region 1, then the algorithm looks for three vertices in regions 3, 7, and 9. This vertex location corresponds to the template $t_5$. If the first vertex of the quadrangle is located in region 2, then the algorithm looks for the remaining vertices in regions 4, 6, and 8. This vertex location corresponds to the template $t_1$. The points most distant from the center of the traffic sign in the desired area correspond to the vertices of the quadrangle.

Quadrangle side equations are used to calculate the deviation of points from the template. The equation of a line is written through the coordinates of the two vertices lying on that line. The distances to these four lines are calculated for each point $(x_i, y_i)$ of contour $D$.

$$d_1(x_i, y_i) = \frac{|(y_1-y_2)(x_i-x_1)+(y_i-y_1)(x_1-x_2)|}{\sqrt{(y_1-y_2)^2+(x_1-x_2)^2}},$$
$$d_2(x_i, y_i) = \frac{|(y_1-y_4)(x_i-x_1)+(y_i-y_1)(x_1-x_4)|}{\sqrt{(y_1-y_4)^2+(x_1-x_4)^2}},$$
$$d_3(x_i, y_i) = \frac{|(y_2-y_3)(x_i-x_2)+(y_i-y_2)(x_2-x_3)|}{\sqrt{(y_2-y_3)^2+(x_2-x_3)^2}}, \qquad (14)$$
$$d_4(x_i, y_i) = \frac{|(y_3-y_4)(x_i-x_3)+(y_i-y_3)(x_3-x_4)|}{\sqrt{(y_3-y_4)^2+(x_3-x_4)^2}}.$$

The contour point must be mapped to the nearest side of the quadrangle template. The deviation from the template for point $(x_i, y_i)$ is equal to the distance to the nearest side of the template.

$$\Delta P_j(x_i, y_i) = \min(d_1(x_i, y_i), d_2(x_i, y_i), d_3(x_i, y_i), d_4(x_i, y_i)).(j = 1, 5) \tag{15}$$

*4.4. Octagon Template ($t_1$)*

Creating an adaptive octagon template is the most difficult task. The traffic sign localization window is divided into eight areas (Figure 12). The algorithm looks for the farthest contour points from the center $(x_c, y_c)$. These points are the vertices of the octagon $(x_k, y_k)$ $(k = 1, \ldots, 8)$.

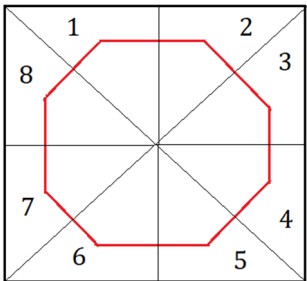

**Figure 12.** Splits the traffic sign localization window to define the vertices of the octagon.

Octagon side equations are also written through vertex coordinates. The distances to the sides of the octagon are calculated for each point $(x_i, y_i)$ of contour $D$.

$$d_k(x_i, y_i) = \frac{|(y_k - y_l)(x_i - x_k) + (y_i - y_k)(x_k - x_l)|}{\sqrt{(y_k - y_l)^2 + (x_k - x_l)^2}},$$
$$l = k (mod\ 8) + 1, k = 1, \ldots, 8. \tag{16}$$

The contour point must be mapped to the nearest side of the octagonal template. The deviation from the template for point $(x_i, y_i)$ is equal to the distance to the nearest side of the template.

$$\Delta P_2(x_i, y_i) = \min_k(d_k(x_i, y_i)).(k = 1 \ldots 8) \tag{17}$$

## 5. Internal Image Recognition

We perform two traffic sign transformations to recognize the internal image. The first transformation removes the edge segment and the entire image in the localization window. The result of this operation is an internal image of the traffic sign. The second transformation scales the internal image to the size of the template. The scaling factor $k$ is equal to the average distance between the vertices of the sign outline to the template size.

For example, for a triangular sign, the scaling factor $k$ is calculated based on the length of the side of the pattern $r$.

$$k = \frac{a_1 + a_2 + a_3}{3r},$$
$$a_1 = \sqrt{(y_1 - y_2)^2 + (x_1 - x_2)^2},$$
$$a_2 = \sqrt{(y_1 - y_3)^2 + (x_1 - x_3)^2},$$
$$a_3 = \sqrt{(y_3 - y_2)^2 + (x_3 - x_2)^2}. \tag{18}$$

$(x_1, y_1)$, $(x_2, y_2)$, and $(x_3, y_3)$ are the coordinates of the vertices for the adaptive template when recognizing the shape of the sign.

Standard internal traffic sign images have two tones. A black picture on a white background is depicted on most traffic signs. Some images have a white picture on a blue

background. The algorithm binarizes the internal image to eliminate unnecessary details. The threshold circuit converts the internal image into a binary matrix $B$. The size $m \times m$ of matrix $B$ is chosen so that it is minimal and the entire traffic sign can be placed in it.

The templates of the internal traffic sign images $T^{(l)}$ are square matrices $m \times m$. $l$ is the template number. Templates $T^{(l)}$ are binary matrices. Matrices $T^{(l)}$ are formed based on reference images of traffic signs. The Hamming distance is used to match the internal image of the traffic sign $B$ and the template $T^{(l)}$.

$$b_l = \sum_{i,j=1}^{n} \left| B_{ij} - T_{ij}^{(l)} \right|. \tag{19}$$

The algorithm compares the internal image with all possible templates. The internal traffic sign image corresponds to pattern $T^{(l)}$ if $b_l$ is minimal. Examples of the algorithm operation to select the edge segment and the internal image are shown in Figure 13.

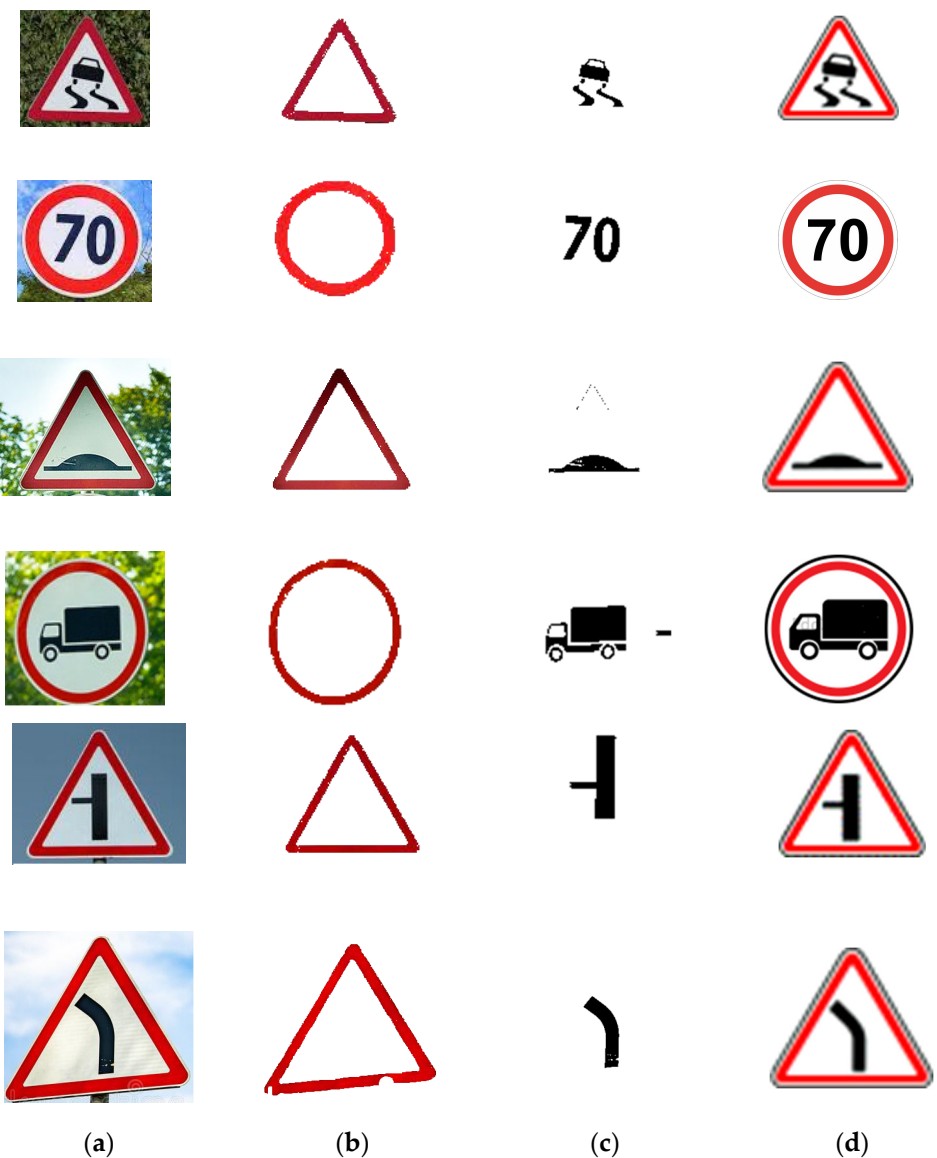

(**a**)  (**b**)  (**c**)  (**d**)

**Figure 13.** Examples of the algorithm operation for selecting an edge segment and an internal image: (**a**) a traffic sign image, (**b**) a traffic sign edge segment, (**c**) an internal traffic sign image, and (**d**) an internal image template.

## 6. Results

We used a regular desktop computer. The software package was implemented in C++. The experiment was carried out on a computer with 12 nuclear processors and 16 GB of RAM. The processing time of one image did not exceed 2 ms. The processing time depended on the type of traffic sign. Some traffic signs were identified only by their shape. The processing time of these was halved. The first-level classifier worked on forming a comparison to the adaptive templates. The efficiency of determining the shape for the sign was 99.8% in the GTSRB collection. The percentage of false positives was 0.4%. Errors in the definition of the shape occur when overlapping the contour of the traffic sign with foreign objects or sharp shadows. Fragmentation of the edge segment occurs in these cases. An incomplete edge segment is mapped to an erroneous adaptive template. The proposed algorithm for sign shape recognition shows resistance to image blurring. This stability allows different filters to be used for image preprocessing.

The color classifier of the edge segment unmistakably works. Its high performance is associated with the structure of traffic signs.

The internal image recognition for traffic signs was also highly efficient. The algorithm correctly determined 99.6% of the images in the GTSRB collection. Errors can occur in low light or with high-angle shooting. Poor illumination requires a careful selection of the image binarization threshold. High-angle shooting effects can be eliminated by geometric transformations of the image. These transformations should include rotating the image by an angle, not just scaling. This question requires more research and is not included in this article.

The algorithm was tested for a collection of real images. The camera on the windshield of the car received pictures of road scenes. The shooting was carried out in various levels of light in the daytime and evening. The total number of signs in the collection was 500. The collection of test images included all possible signs. The frequency of recurrence for the signs in the collection corresponds to the frequency of their occurrence on the roads. The distortion of signs with poor illumination is not related to the shape of the signs and is random in nature. The size of the images ranged from $50 \times 50$ pixels to $200 \times 200$ pixels. The recognition efficiency was 96%. Recognition errors occur when there are objects that cover part of the sign. The algorithm has the greatest difficulty with the overlapping parts of the boundary segment of a traffic sign. Sharp shadows from objects prevent the algorithm from working as well as objects that overlap part of the sign. The algorithm copes with the unevenness of the illumination of the traffic sign image. The segmentation algorithm highlights edge segments with a gradient fill. The threshold template comparison scheme is insensitive to gradient color changes.

Our traffic sign image recognition system uses memory to store templates for internal images. The total number of templates is 132. Each template is 1 KB in size. The total size of the template set is 132 KB. The size of the template base is significantly less than the volume of neural network parameters, which is equal to several megabytes. The template base is easily scalable. The size of one template is small enough that the template base is not large.

## 7. Conclusions

We propose an algorithm for recognizing traffic signs by their images. The proposed algorithm uses additional information about the structure of traffic signs. This information allows to build a hierarchy of traffic sign classes. The advantage of the proposed approach is that it reduces the amount of information to make a decision. Reducing the amount of classifier input is an important factor in the task of recognizing traffic signs. Our algorithm requires fewer computational resources.

Real images for traffic signs are very different from their ideal standards. The main three problems in recognizing the images of traffic signs are the distortion of shapes in the images, color changes in natural light, and overlapping parts of images of signs with foreign objects or shadows from them. Our algorithm solves the problem of distorting the shape of

a sign using adaptable patterns. Geometric transformations correct the internal image of signs and improve the effectiveness of the comparison with templates. The irregularity of the lighting leads to a distortion of the color of the traffic sign. The algorithm works with the color of a traffic sign to determine its shape and type. Uneven illumination changes a uniform fill to a gradient fill. The intensity gradient of the color for the sign can have a complex structure. Our segmentation method contains an adaptive parameter of the pixel color proximity measure. This approach selects gradient fill segments. Objects blocking part of the traffic sign image create the greatest difficulties for the algorithm. The algorithm considers such items as the boundaries of the edge segment. If the edge segment is not defined correctly, then the algorithm does not correctly determine the shape of the traffic sign. The character recognition is incorrect. Foreign objects and contrasting shadows are a common problem for all traffic sign recognition algorithms. Effective methods to overcome this problem do not exist today.

The high accuracy of our algorithm makes it suitable for use in real systems. Its low demand for computing resources allows us to use complex software in mobile laboratories. The high performance of the algorithm and its high efficiency increase the reliability of the input information in decision-making systems of unmanned vehicles and make them more reliable.

**Author Contributions:** Conceptualization, S.V.B.; methodology, S.V.B. and S.Y.B.; software, S.V.B. and E.V.K.; formal analysis, S.Y.B.; data curation, E.V.K.; writing—original draft preparation, S.V.B.; visualization, E.V.K.; project administration, S.V.B. All authors have read and agreed to the published version of the manuscript.

**Funding:** This research was funded by Siberian State Automobile and Highway University.

**Conflicts of Interest:** The authors declare no conflict of interest.

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
