# Peer review of "Hierarchical System for Recognition of Traffic Signs Based on Segmentation of Their Images"

_information, doi:10.3390/info14060335_

Round 1
Reviewer 1 Report
1. The experiments need enhancing.
2. The dataset used in experiments need proving in detail.
3. Extensive experimental results need providing, such as the recognition accuracy of the proposed algorithm.
4. The experimental results should be compared with similar studies.
The English is good.
Author Response
Dear reviewer!
We thank you for your constructive comments. We provide answers to your questions and comments.
- The experiments need enhancing.
Answer: A paragraph has been added to the text of the article.
“The algorithm was tested for a collection of real images. The camera on the windshield of the car received pictures of road scenes. The shooting was carried out in various illuminations in the daytime and evening. The total number of signs in the collection is 500. The collection of test images included all possible sign. The frequency of recurrence for signs in the collection corresponds to the frequency of their occurrence on the roads. Distortion of signs with poor illumination is not related to the shape of signs and is random in nature. The size of the images ranged from 50x50 pixels to 200x200 pixels. Recognition efficiency was 96%. Recognition errors occur when there are objects that cover part of the sign. The algorithm has the greatest difficulty in overlapping part of the boundary segment of the traffic sign. Sharp shadows from objects prevent the algorithm from working as well as objects that overlap part of the sign. The algorithm copes with the unevenness of the illumination of the traffic sign image. The segmentation algorithm highlights edge segments with gradient fill. The threshold template comparison scheme is insensitive to gradient color change.”
- The dataset used in experiments need proving in detail.
Answer: We performed the experiment on a widespread collection used in comparing traffic sign image recognition algorithms. We checked the operation of the algorithm in real images. Our system is currently being used in the road inventory laboratory. We cannot provide more voluminous statistics at the moment.
The problem of sampling data for algorithm testing is the subject of separate study. Image quality requirements are related to the level requirements of the traffic sign recognition system. We are dealing with this problem. However, the task is large enough and requires a separate article. Currently, when compiling samples of images of traffic signs, some restrictions on interference are introduced. These requirements are not explicitly indicated anywhere. The classification of interference that the recognition system copes with leads to the classification of recognition tasks and the classification of recognition systems.
- Extensive experimental results need providing, such as the recognition accuracy of the proposed algorithm.
Answer: The accuracy of recognition in our article is indicated as the efficiency of the algorithm. This parameter is basic to any traffic sign image recognition algorithm. We give the values for the efficiency for various algorithms in the introduction. The effectiveness of our algorithm is given in conclusion.
- The experimental results should be compared with similar studies.
Answer: An overview of the results of similar studies is provided in the introduction. The main characteristic of the algorithm is the efficiency of recognition. This parameter is specified in the article for our algorithm and similar algorithms based on artificial neural networks.
Reviewer 2 Report
A minor revision of the article is given based on the comments below:
You mentioned "Our algorithm requires less computational resources". Can you describe a little more how big the difference is, how much shorter is the recognition time?
Can your algorithm support other traffic sign bases? Did you do any research on other bases?
What if there are more faded traffic signs in the traffic sign database? What if there are traffic signs that are crossed (sprayed) out?
How do you solve the problem if you have two traffic signs in the picture (eg speed limit 50 and no overtaking)?
In conclusion, it would be good to list and then describe the scientific contribution...
Author Response
Dear reviewer!
We thank you for your constructive comments. We provide answers to your questions and comments.
- You mentioned "Our algorithm requires less computational resources". Can you describe a little more how big the difference is, how much shorter is the recognition time?
Answer: A paragraph has been added to the text of the article.
“Our traffic sign image recognition system uses memory to store templates for internal images. The total number of templates is 132. Each template is 1 Kb in size. The total size of the template set is 132 KB. The size of the template base is significantly less than the volume of neural network parameters, which is equal to several megabytes. The template base is easily scalable. The size of one template is small enough that the template base is not large.”
- Can your algorithm support other traffic sign bases? Did you do any research on other bases?
Answer: We performed studies based on GTSRB traffic sign images as well as self-generated images. In the text of the article, a paragraph was added about a selection of characters received on their own.
“The algorithm was tested for a collection of real images. The camera on the windshield of the car received pictures of road scenes. The shooting was carried out in various illuminations in the daytime and evening. The total number of signs in the collection is 500. The collection of test images included all possible sign. The frequency of recurrence for signs in the collection corresponds to the frequency of their occurrence on the roads. Distortion of signs with poor illumination is not related to the shape of signs and is random in nature. The size of the images ranged from 50x50 pixels to 200x200 pixels. Recognition efficiency was 96%. Recognition errors occur when there are objects that cover part of the sign. The algorithm has the greatest difficulty in overlapping part of the boundary segment of the traffic sign. Sharp shadows from objects prevent the algorithm from working as well as objects that overlap part of the sign. The algorithm copes with the unevenness of the illumination of the traffic sign image. The segmentation algorithm highlights edge segments with gradient fill. The threshold template comparison scheme is insensitive to gradient color change.”
- What if there are more faded traffic signs in the traffic sign database? What if there are traffic signs that are crossed (sprayed) out?
Answer: The algorithm is mistaken if there are objects on the image that overlap a significant edge segment or pass through the entire sign. The edge segment and shape of the sign are erroneously determined. No algorithm can handle this problem at this time. If the faded road sign retains a uniform or gradient fill, then the algorithm processes it correctly. If the main color features of the traffic sign are lost completely on the image fragment, then the algorithm is wrong.
- How do you solve the problem if you have two traffic signs in the picture (eg speed limit 50 and no overtaking)?
Answer: This problem is only partly related to our algorithm. We do not solve the problem of localizing road signs. If one of the characters is localized, then the algorithm recognizes this character. The transition to the second sign does not occur during segmentation. A small gap between the signs ensures their separate segmentation.
Reviewer 3 Report
The authors emphasis on developing a hierarchical system based on image segmentation for traffic sign recognition, with the aim of enhancing road safety, is both significant and relevant in the age of intelligent technologies. However, before recommending further action, I would appreciate the views of the authors in respect of the following comments:
-
The authors should provide a step-by-step breakdown of the segmentation process, including pseudo-code, algorithms, or techniques to improve comprehensibility and understanding of their proposed methodology. This will help readers understand the methodology more effectively.
-
The technical soundness of the paper would be enhanced by the inclusion of reasoning behind the chosen segmentation approach and highlighting its advantages over alternative methods. This will add clarity and reinforce the significance of the proposed technique.
-
It is recommended that the authors include detailed information about the dataset used for evaluation, such as size, diversity, and sources, to enhance reproducibility and enable better comparisons with existing studies. This will provide important context and facilitate further research in the field.
-
It is important for the authors to present information about the hardware and software configurations used during experimentation to facilitate replication or further research. In addition, this will help readers understand the technical setup and provide a basis for comparison in future studies.
-
The authors are encouraged to present and discuss the results more comprehensively, going beyond a superficial overview. By delving deeper into the analysis and interpretation of the evaluation metrics, they can provide detailed explanations and valuable insights that will aid readers in understanding and interpreting the findings. This in-depth analysis will allow readers to fully grasp the significance and implications of the results.
-
It would be beneficial for the authors to present how the proposed methodology addresses real-world scenarios, including challenges like occlusion, blur, lighting effects, etc. This will demonstrate the practical relevance and potential applicability of the research. Furthermore, a comprehensive analysis and interpretation of the results, addressing any observed limitations, challenges, and potential sources of error, will provide a more thorough understanding of the system's effectiveness and reliability.
-
It is suggested that the authors should include a thorough discussion of the proposed methodology's applicability and performance in real-world settings. This will help readers assess the research's practical value and potential impact.
Author Response
Dear reviewer!
We thank you for your constructive comments. We provide answers to your questions and comments.
- The authors should provide a step-by-step breakdown of the segmentation process, including pseudo-code, algorithms, or techniques to improve comprehensibility and understanding of their proposed methodology. This will help readers understand the methodology more effectively.
Answer: Our segmentation algorithm is a modification of the usual region growing algorithm. The novelty of the algorithm consists in the method of deciding on including the next pixel in the segment. The decision method is based on modeling the image as a weighted graph and finding the community on this graph. The pseudocode representation of the algorithm is not of interest. Understanding the model is necessary to implement the algorithm. We believe that the model is described in sufficient detail.
- The technical soundness of the paper would be enhanced by the inclusion of reasoning behind the chosen segmentation approach and highlighting its advantages over alternative methods. This will add clarity and reinforce the significance of the proposed technique.
Answer: A paragraph has been added to the text of the article.
“This segmentation algorithm highlights segments that have both a uniform color fill and a gradient color fill. This factor is important in the processing of real traffic signs. The illumination of the sign is never uniform. This results in color gradients in the image of real traffic signs. Our algorithm has a quadratic complexity. The speed of the algorithm is large enough for use in the designed system.”
- It is recommended that the authors include detailed information about the dataset used for evaluation, such as size, diversity, and sources, to enhance reproducibility and enable better comparisons with existing studies. This will provide important context and facilitate further research in the field.
Answer: A paragraph has been added to the text of the article.
“The collection of test images included all possible sign. The frequency of recurrence for signs in the collection corresponds to the frequency of their occurrence on the roads. Distortion of signs with poor illumination is not related to the shape of signs and is random in nature. The size of the images ranged from 50x50 pixels to 200x200 pixels.”
- It is important for the authors to present information about the hardware and software configurations used during experimentation to facilitate replication or further research. In addition, this will help readers understand the technical setup and provide a basis for comparison in future studies.
Answer: A paragraph has been added to the text of the article.
“The software package is implemented in C++. The experiment was carried out on a computer with 12 nuclear processors and 16 GB of RAM. The processing time of one image did not exceed 2 ms. The processing time depends on the type of traffic sign. Some traffic signs are identified only by their shape. The processing time of such characters is halved.”
- The authors are encouraged to present and discuss the results more comprehensively, going beyond a superficial overview. By delving deeper into the analysis and interpretation of the evaluation metrics, they can provide detailed explanations and valuable insights that will aid readers in understanding and interpreting the findings. This in-depth analysis will allow readers to fully grasp the significance and implications of the results.
It would be beneficial for the authors to present how the proposed methodology addresses real-world scenarios, including challenges like occlusion, blur, lighting effects, etc. This will demonstrate the practical relevance and potential applicability of the research. Furthermore, a comprehensive analysis and interpretation of the results, addressing any observed limitations, challenges, and potential sources of error, will provide a more thorough understanding of the system's effectiveness and reliability.
Answer: A paragraph has been added to the text of the article.
“Real images for traffic signs are very different from their ideal standards. The main three problems in recognizing the image of traffic signs are distortion of the shape in the image, color change in natural light and overlapping part of the image of the sign with foreign objects or shadow from them. Our algorithm solves the problem of distorting the shape of a sign using adaptable patterns. Geometric transformations correct the internal image of signs and improve the effectiveness of comparison with templates. The irregularity of the lighting leads to distortion of the color of the traffic sign. The algorithm works with the color of a traffic sign to determine its shape and type. Uneven illumination changes a uniform fill to a gradient fill. The intensity gradient of the color for the sign can have a complex structure. Our segmentation method contains an adaptive parameter of the pixel color proximity measure. This approach selects gradient fill segments. Objects blocking part of the traffic sign image create the greatest difficulties for the algorithm. The algorithm considers such items as the boundaries of the edge segment. If the edge segment is not defined correctly, then the algorithm does not correctly determine the shape of the traffic sign. The character recognition is incorrect. Foreign objects and contrasting shadows are a common problem for all traffic sign recognition algorithms. Effective methods to overcome this problem do not exist today.”
- It is suggested that the authors should include a thorough discussion of the proposed methodology's applicability and performance in real-world settings. This will help readers assess the research's practical value and potential impact.
Answer: A paragraph has been added to the text of the article.
“The algorithm was tested for a collection of real images. The camera on the windshield of the car received pictures of road scenes. The shooting was carried out in various illuminations in the daytime and evening. The total number of signs in the collection is 500. The collection of test images included all possible sign. The frequency of recurrence for signs in the collection corresponds to the frequency of their occurrence on the roads. Distortion of signs with poor illumination is not related to the shape of signs and is random in nature. The size of the images ranged from 50x50 pixels to 200x200 pixels. Recognition efficiency was 96%. Recognition errors occur when there are objects that cover part of the sign. The algorithm has the greatest difficulty in overlapping part of the boundary segment of the traffic sign. Sharp shadows from objects prevent the algorithm from working as well as objects that overlap part of the sign. The algorithm copes with the unevenness of the illumination of the traffic sign image. The segmentation algorithm highlights edge segments with gradient fill. The threshold template comparison scheme is insensitive to gradient color change.”
Round 2
Reviewer 3 Report
Prior to recommending further action, the authors should address the following comments to enhance the clarity, validity, and applicability of the paper.
1. To enhance the overall structure and coherence of the paper, we highly recommend that the authors establish separate sections for Discussion and Results rather than combining them into the conclusion section. This separation will provide a clearer organization of the paper and make it easier for readers to navigate through the content.
2. On page 6, the authors mention that their algorithm has quadratic complexity and claim that its speed is sufficient for use in the designed system. However, it is important to provide mathematical proof or equations to support this claim. Additionally, it would be beneficial to address the potential challenges associated with quadratic complexity, particularly when dealing with significantly large datasets. As the input size increases, the execution time can quickly become impractical or inefficient.
3. The authors state that they have achieved a 99.8% accuracy rate in identifying the shape of signs and a 99.6% accuracy rate in internal image recognition on the GTSRB dataset. It is crucial for the authors to explain how these high accuracy rates in shape identification and internal image recognition would improve the overall reliability and performance of TSR (Traffic Sign Recognition) systems. Providing insights into the practical implications and benefits of such accuracy rates will strengthen the paper's argument.
4. Real-world testing and evaluation of the system's performance beyond the GTSRB dataset are necessary to accurately determine the model's actual performance. It is important to include results from testing the proposed algorithm on diverse real-world datasets, as this will provide a more comprehensive evaluation of its effectiveness in handling various lighting conditions, weather conditions, occlusions, and other factors encountered in real-world road scenes. This will enhance the paper's credibility and applicability.
5. To provide a more comprehensive analysis, we recommend including a comparative analysis with other state-of-the-art (SOTA) methodologies. This analysis will allow readers to understand the strengths and weaknesses of the proposed algorithm in relation to existing approaches. Including a broader comparison will provide a more well-rounded assessment of the proposed algorithm's performance.
6. On page 14, the authors mention that the experiment was conducted on a computer with 12 nuclear processors and 16 GB of RAM. While this information is relevant, it would be beneficial to include additional details about the hardware specification. These details will provide a clearer understanding of the computational resources required to run the algorithm effectively.
Author Response
Dear reviewer!
We thank you for your constructive comments. We provide answers to your questions and comments.
- To enhance the overall structure and coherence of the paper, we highly recommend that the authors establish separate sections for Discussion and Results rather than combining them into the conclusion section. This separation will provide a clearer organization of the paper and make it easier for readers to navigate through the content.
Answer: The "Results" section has been added to the article.
- On page 6, the authors mention that their algorithm has quadratic complexity and claim that its speed is sufficient for use in the designed system. However, it is important to provide mathematical proof or equations to support this claim. Additionally, it would be beneficial to address the potential challenges associated with quadratic complexity, particularly when dealing with significantly large datasets. As the input size increases, the execution time can quickly become impractical or inefficient.
Answer: A paragraph has been added to the text of the article (line 183).
“ The segmentation algorithm is based on the method of growing areas. The method of growing areas has a quadratic labor intensity. Each iteration of the algorithm performs a number of operations not exceeding the number of image points. The total number of iterations does not exceed the number of image points. The speed of the algorithm is large enough for use in the designed system. Размеры входных изображений алгоритма не превышают 200x200 пикселей.”
- The authors state that they have achieved a 99.8% accuracy rate in identifying the shape of signs and a 99.6% accuracy rate in internal image recognition on the GTSRB dataset. It is crucial for the authors to explain how these high accuracy rates in shape identification and internal image recognition would improve the overall reliability and performance of TSR (Traffic Sign Recognition) systems. Providing insights into the practical implications and benefits of such accuracy rates will strengthen the paper's argument.
Answer: A paragraph has been added to the text of the article.
“The high accuracy of the algorithm makes it suitable for use in real systems. Low demand for computing resources allows to use the software complex in mobile laboratories. The high performance of the algorithm, together with high efficiency, increase the reliability of input information in decision-making systems of unmanned vehicles and make them more reliable.”
- Real-world testing and evaluation of the system's performance beyond the GTSRB dataset are necessary to accurately determine the model's actual performance. It is important to include results from testing the proposed algorithm on diverse real-world datasets, as this will provide a more comprehensive evaluation of its effectiveness in handling various lighting conditions, weather conditions, occlusions, and other factors encountered in real-world road scenes. This will enhance the paper's credibility and applicability.
Answer: Test results on real road scenes are in the article (lines 427).
- To provide a more comprehensive analysis, we recommend including a comparative analysis with other state-of-the-art (SOTA) methodologies. This analysis will allow readers to understand the strengths and weaknesses of the proposed algorithm in relation to existing approaches. Including a broader comparison will provide a more well-rounded assessment of the proposed algorithm's performance.
Answer: We cited the effectiveness of other algorithms in the introduction. The main difference between our algorithm and others is indicated in the conclusion. A complete answer to your question requires writing a large review article.
- On page 14, the authors mention that the experiment was conducted on a computer with 12 nuclear processorsand 16 GB of RAM. While this information is relevant, it would be beneficial to include additional details about the hardware specification. These details will provide a clearer understanding of the computational resources required to run the algorithm effectively.
Answer: There was a typo in the sentence. The correct sentence is as follows:
“The experiment was carried out on a computer with 12 nuclear processor and 16 GB of RAM.”
We used a regular desktop computer. Our algorithm does not require special equipment.